# Multifunctional Amyloids in the Biology of Gram-Positive Bacteria

**DOI:** 10.3390/microorganisms8122020

**Published:** 2020-12-17

**Authors:** Ana Álvarez-Mena, Jesús Cámara-Almirón, Antonio de Vicente, Diego Romero

**Affiliations:** Instituto de Hortofruticultura Subtropical y Mediterránea “La Mayora”—Departamento de Microbiología, Universidad de Málaga, Bulevar Louis Pasteur, 31 (Campus Universitario de Teatinos), 29071 Malaga, Spain; alvarezmena@uma.es (A.Á.-M.); jesus_camara@uma.es (J.C.-A.); adevicente@uma.es (A.d.V.)

**Keywords:** functional amyloids, gram-postive bacteria, *Bacillus*, microbial ecology, biofilm, pathogenesis

## Abstract

Since they were discovered, amyloids have proven to be versatile proteins able to participate in a variety of cellular functions across all kingdoms of life. This multitask trait seems to reside in their ability to coexist as monomers, aggregates or fibrillar entities, with morphological and biochemical peculiarities. It is precisely this common molecular behaviour that allows amyloids to cross react with one another, triggering heterologous aggregation. In bacteria, many of these functional amyloids are devoted to the assembly of biofilms by organizing the matrix scaffold that keeps cells together. However, consistent with their notion of multifunctional proteins, functional amyloids participate in other biological roles within the same organisms, and emerging unprecedented functions are being discovered. In this review, we focus on functional amyloids reported in gram-positive bacteria, which are diverse in their assembly mechanisms and remarkably specific in their biological functions that they perform. Finally, we consider cross-seeding between functional amyloids as an emerging theme in interspecies interactions that contributes to the diversification of bacterial biology.

## 1. Introduction

The term “amyloid” has been historically associated with misfolded proteins with a tendency to accumulate in neural tissue, causing a broad range of neurological disorders known as proteopathies or protein misfolding diseases, such as Alzheimer’s disease or Parkinson’s disease [1]. The histological manifestation of these pathological conditions is, indeed, the amyloid deposits that were first observed by pathologist Rudolph Virchow in 1854 [2]. These deposits resembled carbohydrate starches when stained with iodine compounds, which in Latin are called *amylum* and hence the term “amyloid”. During the 1960s and 1970s it was demonstrated that some of these disorders can be experimentally transmitted to animals and, in some conditions, to humans. Since then, the study of the transmission of these protein misfolded diseases has been key to understand the properties of amyloid proteins. Such is the case of Alzheimer’s disease [3] in which the use of contaminated surgical instruments and blood products in patients with no prior history of neurological disorders led to changes in the brain that resemble those found in patients with Alzheimer’s disease, due to the presence of Aβ seeds adhered to metal surfaces that can resist the inactivation by formaldehyde or other physicochemical treatments [4].

Currently, amyloids are not simply misfolded proteins; they are widely distributed from prokaryotic to eukaryotic and are implicated in a broad range of biological processes, such as protection, interaction with abiotic and biotic surfaces and detoxification [5,6,7,8,9,10,11,12,13]. For these reasons, some amyloid proteins have been termed “functional amyloids”, which refers to their ability to perform useful biological functions for the organism.

One of the defining features of amyloid proteins is the differences in their amino acid sequence and, despite this difference, the strongly conserved quaternary structure of cross-β strands [14]. This well-ordered architecture is formed by oligomers that adopt a β-sheet structure noncovalently bound by hydrogen bonds. These proteins have the ability to self-assemble in a nucleation-dependent process from monomers to oligomers that ultimately lead to fibres with a diameter of a few nanometres and with a length from tens of nanometres to several micrometres. This β-sheet enrichment confers tinctorial properties of amyloid proteins, and can be stained by dyes such as Congo red or thioflavin T (ThT) [15,16]. Moreover, the canonical, ordered, cross-β architecture can be identified by X-ray diffraction. Nonetheless, detection by nuclear magnetic resonance (NMR), Fourier transform infrared spectroscopy (FTIR) or even paramagnetic resonance profiles are other alternatives to study the structure of amyloid proteins [17]. 

An interesting trait shown by some amyloids is their ability to interact with one another in a process known as cross-seeding, which has attracted interest in recent years due to the interaction of different amyloids between different species. In fact, it has been shown that the cross-seeding between bacterial and human amyloids can lead to an enhancement of amyloidosis related to neurodegenerative disorders. Further studies are needed to determine, in detail, the effects of bacterial amyloids in human deposits and in the host immune response [18].

The conserved cross-β structure gives amyloid proteins characteristic biophysical properties, such as insolubility and high resistance to different physicochemical aggressions, such as proteolysis, detergent treatments, and chemical or thermal denaturation [19]. Due to the high stability and robustness exhibited by this family of proteins, amyloids are frequently compared to steel in terms of strength [20,21]. 

In prokaryotes, functional amyloids have been identified in *Proteobacteria*, *Bacteroidetes*, *Chloroflexi*, *Actinobacteria*, and *Firmicutes* [22,23,24]. The most studied and first identified bacterial amyloid was curli from *Escherichia coli*, which is implicated in a variety of physiological functions [8,25] and that has been later identified in strains of *Salmonella*, *Citrobacter* and *Enterobacter* [26]. Since then, many amyloids have been reported as versatile proteins that are mainly involved in not only biofilm formation and adhesion to surfaces but also cytotoxicity, virulence, plasmid replication and reproduction [6,7,8,9,11,27,28,29,30,31,32,33]. Bacterial biofilms are communities of a single or multiple species of microorganisms attached to a surface formed in response to a myriad of signals that trigger a process of cellular differentiation and secretion of a multifunctional extracellular matrix (ECM). Protecting cells from environmental stress (antimicrobials, host defences, UV radiation, desiccation, mechanical stress, and phage predation), providing stability to the biofilm architecture and regulating the flow of nutrients and signalling molecules are among the essential functions covered by the ECM [34,35,36]. The ECM is mainly composed of exopolysaccharides, extracellular DNA, lipids, and proteins, but the proportions of these biomolecules vary depending on the species. Due to their chemical features, amyloids are members of the ECM, participate in adhesion to biotic or abiotic surfaces, provide robustness to biofilms, and compose a structural scaffold that serves to build the microbial community [10,11,37,38]. However, the unique features of amyloid proteins have captivated the interest of different disciplines beyond pathology and bacteriology, such as nanomaterial engineering and other biotechnological applications [39,40]. Emerging pharmacological studies are focused on the inhibition of amyloidogenesis as an antibacterial or antivirulence therapy, although some compounds promote the development of neurodegenerative diseases in humans, so dedicated attention should be maintained [41].

In this review article, we present the current knowledge and latest findings of different bacterial amyloid systems, with particular emphasis on gram-positive bacteria, their contribution to bacterial multicellularity and microbe-host interactions, and how the process of cross-seeding between different bacterial amyloids can diversify their functionality in microbial biology.

## 2. The Hydrophobic Layers of *Streptomyces coelicolor*: Chaplins and Rodlins

One of the first functional amyloids reported in gram-positive bacteria was in *Streptomyces coelicolor*, a soil-dwelling bacterium with a lifecycle similar to that of filamentous fungi, in which a submerged mycelium emerges and aerial hyphae are developed to finally septate into chains of spores [28,42]. After being dispersed by insects or wind, the spores can germinate to form a new vegetative mycelium. The surface of aerial hyphae is extremely hydrophobic compared to the submerged hyphae, and three functional amyloids are involved in the formation of these structures: chaplins, rodlins and SapB [43,44].

Chaplins (coelicolor hydrophobic aerial proteins) are a family composed of eight secreted proteins (ChpA-H), but only the *chpC*, *chpE* and *chpH* chaplin genes are conserved in all *Streptomyces* species [45,46,47]. Except for *chpB*, all chaplin genes are encoded in the core region of the lineal chromosome of *S. coelicolor* [42]. Strains with the deletion of individual or pairs of chaplin genes do not show phenotypic changes, so a possible redundancy in function could exist between these proteins [28]. In fact, a “minimal chaplin strain”, encoding only the three conserved genes mentioned above, exhibits a wild-type-like phenotype with a robust aerial mycelium [48]. Furthermore, it is known by transcriptomic analysis that there is a temporal differentiation in the expression pattern of the different chaplin genes, in which *chpC*, *chpE* and *chpH* are expressed in an early stage of aerial hyphal formation, and *chpE* and *chpH* are induced during the later stages of the developmental cycle [28,42]. Streptomycetes are able to adhere to different surfaces, with the ability to grow as an invasive form or to establish an infection in the case of pathogenic strains. Fimbrial structures are implicated in attachment, and in *Streptomyces coelicolor* protrusions emerging along the surface of the hyphae are chaplin-dependent, suggesting that fimbriae might be composed of or related to these functional amyloids [49].

All chaplins possess a conserved region called the chaplin domain, are highly hydrophobic and contain three conserved GN motifs. Additionally, each amino acid sequence possesses two cysteine residues (except for ChpE) that form intramolecular disulfide bonds that, in the case of ChpH, are essential for the development of aerial hyphae and for the building of the rodlet ultrastructure [48]. The chaplin family is divided into two groups, depending on the number of these chaplin domains: (i) the long chaplins (ChpA-C), which have two chaplin domains separated by 35 amino acids and a C-terminal signal recognized by a sortase for covalent attachment to the cell wall; and (ii) the short chaplins (ChpD-H), which contain a single chaplin domain [28,50,51]. A model for chaplin assembly proposed by Elliot, M.A. and colleagues suggested that long chaplins have a role as a scaffold for the proper polymerization and attachment of short chaplins to the cell wall, thereby forming, in conjunction, a hydrophobic surface that provides protection from and reduction in the surface tension during the development of aerial hyphae [28]. Given the amyloid nature of chaplins, these proteins are enriched in β-sheet secondary structures and are highly insoluble and can be stained with the amyloid-specific dyes Congo red and ThT [42,49]. ChpH is one of the most studied chaplin proteins due to its presence in all *Streptomyces* species. In the ChpH sequence, there are two regions that are described as amyloidogenic domains and that are responsible for the assembly process in other amyloids [52,53]. In vivo studies have shown that both domains are indispensable for the development of aerial hyphae, but only the region located near the C-terminal end participates in the formation of the rodlet surface [54].

Rodlins (RdlA and RdlB) play a similar role as fungal hydrophobins and are able to polymerize into a structure that resembles a basketwork of paired rodlets (also known as the rodlet layer) on the surface of aerial hyphae and spores. These proteins are known to be involved in the attachment of hyphae to hydrophobic surfaces [55]. This ordered structure is larger than the chaplin fibres, which are organized into pairs along the surface by the activity of RdlA and RdlB [55,56]. Studies regarding the surfactant activity of these proteins, however, demonstrated that rodlins are less effective in lowering the surface tension than chaplins [57]. Rodlins are poorly biochemically characterized, given the impossibility of separating them using the established trifluoroacetic acid extraction procedure [56]. Despite the similarity between the two Rdl proteins (>90%), each one has a unique functional role, and therefore, both are essential in the assembly process [57]. Indeed, both proteins differ by only a few amino acids located in their N-terminal domain, which is the reason for the lower net charge and the higher hydrophobicity of RdlB than RdlA. RdlB presents a higher tendency to form amyloid-like fibres in vitro; however, RdlA barely forms a few irregular amorphous aggregates [57].

As mentioned before, chaplin and rodlin fibres are located on the surface of aerial hyphae and spores, assemble into a paired-rod ultrastructure called the rodlet layer [42,56], provide protection against desiccation and provide support as scaffolds due to the high level of hydrophobicity exhibited by these amyloid proteins. The mechanism by which chaplins and rodlins cooperate together is crucial to understanding how the rodlet layer is formed.

Finally, SapB (spore-associated protein B) is described as a “lantibiotic-like” molecule, although antibiotic activity has not been reported for this molecule [58], and acts as a surfactant under high osmolarity [59]. Under these environmental conditions, when releasing the hyphae into the air is difficult due to a lower turgor pressure, SapB is hypothesized to work cooperatively with chaplins in reducing their assembly at the air-liquid interface and therefore prevents the formation of a rigid membrane that increases the time of high surface activity [53,60]. Peptides that are homologous to SapB have been identified in other *Streptomyces* species [44,61,62].

## 3. Multifunctional Amyloids in Staphylococci: Phenol Soluble Modulins (PSMs), BapC, and SuhB

The opportunistic pathogen *Staphylococcus aureus* is a resident bacterium of the skin or mucosa of mammals, producing, in some cases, severe infections and septicemia, and *Staphylococcus epidermidis*, which is often found in different medical devices or indwelling catheters, is known for its tendency to form biofilms that are resistant to the host immune response, chemotherapy and disinfectants [63]. In 1999, phenol soluble modulin (PSM) peptides were described by Seymour Klebanoff as a pro-inflammatory “complex” in the phenol phase during hot phenol extraction [64]. To date, PSMs, which are peptides with notable surfactant properties, have been extensively studied in these two bacterial species [65,66]. It is known that each staphylococcal species possesses a specific cluster of PSMs, and some of them, but not all, share sequence homology to some extent. This difference precisely could reflect the diversity in the lifestyles of *S. aureus* and *S. epidermidis* [64,67,68,69,70]. Indeed, it has been demonstrated that the expression of PSMs is higher in pathogenic methicillin-resistant strains of *S. aureus* than in strains that have a non-pathogenic lifestyle [69,71], but further studies are needed to better establish a possible correlation between PSM peptides and virulence [72].

PSM peptides can be classified depending on their length. The smallest peptides are α-type PSMα1–4 in *S. aureus*, PSMα, PSMδ and PSMε in *S. epidermidis* and δ-toxin in both *S. aureus* and *S. epidermidis*. This group of peptides is characteristically composed of 20–25 amino acids with either a positive or neutral net charge. The longer peptides, which are 44 amino acids in length, are PSMβ1 and PSMβ2 from *S. aureus* and *S. epidermidis*, and they have a negative net charge [64,67,70]. In *S. aureus*, the genes that encode the PSM peptides are located in three different parts of the core genome: four peptides in the α-operon (psmα1–4), two peptides in the β-operon (psmβ1–2) and one peptide in the δ-toxin locus that is encoded in the region of the regulatory RNA RNAIII [64,69,73]. Similar to other staphylococcal exotoxins, the global regulator Agr (accessory gene regulator) enhances PSM expression and production at a high cell density, thereby promoting virulence [69,74]. PSM peptides have antimicrobial activity against niche bacteria [72,75], and they are also implicated in having pathogenicity, hindering the immune response and lysing host cells [76,77]. Furthermore, these surfactant-like molecules act as functional amyloids in the development of biofilms, giving physical support and stability to the scaffold [11,78]. Despite its sequence similarity, not all PSM peptides form ordered amyloid structures, and the PSMα1 and PSMα4 peptides are described to be the main contributors to fibrillar function, showing a high tendency to self-assembly into cross-β fibres [11,79,80], conferring essential stability to biofilms.

The most studied PSM peptide is the PSMα3 peptide, which exhibits the highest cytotoxic and lytic behaviours and contributes to all functions described for PSMs, from biofilm formation to immune activation [66,76,81]. This peptide maintains an α-helical conformation in solution and in elongated fibres [82], which has been confirmed by X-ray diffraction analysis. PSMα3 was the first functional amyloid whose structure in the fibrillar form was studied at atomic resolution, showing that the α-helices were perpendicular to the fibril axis and stack into sheets that run parallel to the fibril axis. The differences in the secondary and quaternary structures between PSMα1 and PSMα4 fibres, which fold into a canonical cross-β structure, and the PSMα3 fibres, suggest a functional specificity.

As mentioned before, some PSMs are able to lyse a wide range of cell types, such as bone cells, monocytes and erythrocytes, by a mechanism that implies the perturbation of the target membrane in a receptor-independent manner [69,76,83,84,85]. Within the PSM group, the α-type exhibits a greater cytolytic activity than the β-type. In *S. aureus* and *S. epidermidis*, PSMα3 and PSMδ, respectively, are the most cytolytic PSM peptides, displaying cytotoxicity in the micromolar range. The antimicrobial peptides of *S. aureus*, such as the β-toxin or PVL, intensify the cytotoxicity of PSMs, suggesting a synergy between functional amyloids and other molecules [86,87]. PSMα3 is a virulence factor of *S. aureus* that is toxic to T-cells or human embryonic kidney 293 (HEK293) cells [82] but is ineffective against *Bacillus subtilis, Pseudomonas aeruginosa, Micrococcus luteus* or *Streptococcus pyrogenes* bacterial cells [78], indicating a significant degree of specificity. A modified peptide of PSMα3 was designed, in which the phenylalanine located at position 3 was replaced by an alanine. This version of PSMα3 was unable to self-assemble into fibres and was conceived with the aim of elucidating the role of the fibrils in the biological activity and virulence of PSMα3. Surprisingly, this modified peptide exhibited a gain of function, becoming lethal to *B. subtilis* cells but not to *P. aeruginosa* [78]. A study performed by simulating mammalian and bacterial membranes showed that PSMα3 interacts differently with the lipid bilayer depending on the lipid composition, which might provide an additional explanation for the differential cytotoxic behaviour of this peptide [88]. The role of PSMα3 fibrils is not completely understood, but recent work suggests that the positive charges and the cross-α structure are involved in its cytotoxicity, which provokes membrane disruption [69,82].

Apart from PSM peptides, another functional amyloid called Bap (biofilm-associated protein) is also present in the staphylococcal biofilm matrix, where it functions as a surface adhesin. Bap was first identified in bovine mastitis caused by the *S. aureus* strain V329 [89] and is absent in human isolates; therefore, it is considered a host-specific pathogenic factor. Bap is a high molecular weight protein (2276 amino acids in length), is located on the bacterial surface, and has multiple domains along its sequence [90]. Bap is implicated in the initial adhesion to abiotic surfaces, intercellular adhesion and bacteria-host interactions, but the exact function of Bap in biofilm development is still unclear [91]. Moreover, this protein presents some similarities with the well-known curli of *E. coli*: cell surface location, multiple tandem repetitions in the carboxyl domain, coordinated synthesis with exopolysaccharide and biofilm formation by interaction with surfaces or cells [92]. Bap is expressed continuously during the bacterial growth curve [93], and the protein is processed by the cell to an insoluble amyloid-like aggregate that forms at a low pH and in a manner dependent on calcium concentration. At high calcium levels, metal-coordinated Bap is more stable, and the N-terminal domain does not assemble, inhibiting the aggregation between bacteria and, therefore, inhibiting biofilm formation, demonstrating the sensorial role of this protein in sensing the environmental conditions that affect biofilm formation.

Independent of Bap, a transposon mutagenesis study identified the protein SuhB as an essential factor for poly-N-acetylglucosamine-independent biofilm formation in *S. aureus*. Studies in *E. coli* [94], *P. aeruginosa*, *Pseudomonas putida* [95], *Burkholderia cepacia* [96], and others [97], have demonstrated the role of SuhB as a regulator of diverse biological functions, including protein secretion, expression of virulence factors, antibiotic resistance, exopolysaccharide production and biofilm development; however, the specific role in *S. aureus* biofilm formation remains unclear. Macroscopic fibres of SuhB have been observed by electron microscopy, and spectroscopic and structural analysis showed that the fibres formed by SuhB exhibited amyloid properties. Furthermore, this protein is able to adhere to the cell surface, indicating its possible role as a component of the biofilm matrix in *S. aureus* [98].

## 4. Functional Amyloids in the Plaque: *Streptococcus mutans* Adhesin P1, Wall Associated Protein A (WapA) and Secreted Protein SMU_630

*Streptococcus mutans* is capable of forming biofilms to survive and persist in dental plaque, causing human dental caries, also known as cavities. In addition, this gram-positive bacterium has been identified as a causative agent of infectious endocarditis. The tooth surface is covered by a salivary pellicle where *S. mutans* sticks to the salivary agglutinin glycoprotein (SAG) complex, which is mainly composed of the scavenger receptor gp340/PMBT1, host cell matrix proteins and other bacteria [99,100]. Binding to SAG is mediated by cell surface adhesin P1 (also named AgI/II, PAc, SpaP or antigen B), but the molecular mechanism of this interaction remains unclear. P1 is a multifunctional protein able to interact with the eukaryotic extracellular matrix proteins collagen and fibronectin, and it is also involved in the interactions between cells [101,102,103,104,105]. Adhesin P1 is a large protein (185 kDa) with multiple domains: A secretion signal peptide, an N-terminal region, three alanine (A)-rich repetitions, a region called variable (which differs between strains), three proline (P)-rich repetitions, a C-terminal region composed of three domains (C1–3) and, at the end, there is an LPxTG recognition motif for the covalent attachment of the protein to the cell wall by the action of the transpeptidase SrtA [106,107]. The protein folds in an unusual tertiary structure [108,109,110], in which the A and P regions form an α-helix that entwines between them, forming a long narrow stalk. The variable part forms a β-sandwich between two sheets, and the C-terminal domain folds into similar structures. Therefore, the global tertiary structure is an extended stalk with a globular domain containing β-sheets. The C-terminal region of the protein can be found in two versions: as part of the full-length protein, in which it is anchored to the cell wall of *S. mutans* or as a single polypeptide that is covalently linked with the whole protein [111]. In electron microscopy studies of thin sections of *S. mutans* cells, adhesin is seen as a layer of fibres that emerge from the cell wall [112]. Upon heterologous expression, the P1 protein is able to fibrillate into fibres with amyloid properties [29], and it has been recently reported that the C123 domain of P1, also referred to as antigen II (AgII), contains the amyloid forming moiety [113].

The fact that biofilms of the *S. mutans* strain that is deficient in adhesin P1 still exhibits green birefringence under Congo red staining suggests the presence of additional proteins with an amyloid nature [29]. Indeed, two additional functional amyloids have been found in this microorganism: WapA [114] and the poorly characterized protein SMU-63c [113]. WapA is a polypeptide that occurs naturally as a result of the truncation of antigen A [115,116], and it is similar to P1 when it is processed by the transpeptidase SrtA. It is involved in some functions, such as binding to collagen, affecting chain length and assembling biofilms [115,117,118]. To date, the crystal structure of WapA is not known. The ability of the C123 domain, which is present in the three functional amyloids, to assemble into fibres has been analysed [113], and shows that in the aggregated state, this domain forms a broad range of amyloid-like structures, from smaller amyloid fibrils (width of 7–12 nm and length of 15 nm to several micrometres) to higher structures (width of 90–150 nm). Additionally, the amyloid inhibitors epigallocatechin-3-gallate or the benzoquinone derivative AA-861 inhibited the in vitro amyloid formation of the C123 domain and WapA and impaired biofilm development by a mechanism dependent on the three amyloid proteins. It is known that WapA and P1 act during the initial stages of biofilm formation; in contrast, SMU_63c is able to form fibres only under acidic conditions, suggesting the importance of the environmental pH in the different regulation of these three *S. mutans* proteins.

Some questions remain unanswered about the functions of each functional amyloid in *S. mutans* biology, and more research is needed to fully understand this system and the possible interactions between these functional amyloids.

## 5. Functional Amyloids in *Listeria monocytogenes*

This microorganism is a food-borne intracellular pathogen and the causative agent of listeriosis, an infectious disease that can cause severe symptoms such as meningitis or encephalitis and that, eventually, can lead to death [119]. Biofilms play an important role in the survival of this microorganism in the food processing industry, in which this microorganism is able to attach and proliferate over different surfaces [120]. Genome analysis of *L. monocytogenes* EGD revealed an ORF (Lmo0435, designated as BapL) that showed a high percentage of similarity with Bap of *S. aureus* or Esp from *E. faecalis* and that contributed to the attachment of *L. monocytogenes* EGD to different inert surfaces [121]. Interestingly, these two abovementioned proteins have been described as functional amyloids, showing all their canonical properties, and are important for the structuring of the extracellular matrix and multicellularity in response to different environmental conditions [93,122]. However, the amyloid nature of BapL remains to be demonstrated.

A paradigm example of the role of functional amyloids in pathogenic interactions with hosts is the toxin listeriolysin O (LLO). This protein is part of a family of toxins known as cholesterol-dependent cytolysins that are produced by different groups of bacterial pathogens, including *Firmicutes*, *Actinobacteria* and *Proteobacteria*, and that use cholesterol as a receptor [123]. LLO is required for the virulence of most of the strains of *L. monocytogenes*, and it is produced and secreted as a monomer that, upon contact with biological membranes, oligomerizes, forming a pre-pore structure of several units. After binding to the membrane, the α-helical regions of the monomers are refolded into beta-hairpins that are inserted into the membrane, forming a beta-barrel pore [124]. Remarkably, LLO exhibits pH-dependent behaviour that provides the pathogen with an escape strategy from eukaryotic phagolysosomes, ensuring the survival of the microorganism in the host cell and enabling infection. The optimal pH and temperature for the activity of LLO and pore formation is acidic pH (5.5) and 37 °C, which are conditions that resemble those found in human phagolysosomes. Under these environmental conditions, the protein is stable as soluble dimers become functional when they are in contact with the cell membrane, as this contact triggers their pore-forming activity [123]. However, under more alkaline conditions, such as at neutral pH found in the cell cytosol, LLO is inactive and forms amorphous aggregates that have many of the properties of amyloid proteins [125] (Figure 1). In this particular case, the amyloid state has evolved as a mechanism to carefully control the rate of active toxin that is dependent on the surrounding physiological conditions and therefore can control the entry and survival of the pathogen within the eukaryotic host. Apart from this essential function in pathogenesis, LLO seems to contribute to the internalization of this microorganism into host cells, is able to cause apoptosis, limits ROS production in the phagosome and induces mitochondrial fragmentation [126], making this toxin an example of the versatility of functional amyloids.

## 6. The TasA Amyloid System in *Bacillus* spp.

One of the best characterized amyloid systems is the one found in the soil-dwelling bacterium *Bacillus subtilis*, which has long been studied to describe the molecular mechanisms that lead to biofilm formation and other microbial processes [127]. The major protein component of the extracellular matrix [128] and the main amyloid component of this system is TasA (translocation-dependent antimicrobial spore component) due to the broad-spectrum antimicrobial properties reported in the pioneering studies [129]. Later, it was demonstrated that TasA is able to self-assemble into amyloid aggregates and fibres that serve as the structural scaffold of the extracellular matrix in *B. subtilis* biofilms [10]. These TasA amyloid fibres in *B. subtilis* are also composed of the accessory protein TapA (TasA anchoring and assembly protein), which participates in the assembly process by enhancing the polymerization of TasA subunits and anchoring the fibres to the cell surface in vivo [130]. Both proteins are essential for multicellularity and ECM production in *B. subtilis*, given that the absence of either of these proteins leads to defects in pellicle formation or colony architecture.

Structural studies have undoubtedly confirmed the amyloid nature of TasA, which transitions from a globular state to the characteristic robust fibres found in these multicellular communities [131]. Indeed, the existence of an amyloid core within the TasA sequence has been determined, indicating that not all the residues present in the protein are involved in the folding of the amyloid structure of TasA [132] (Figure 2A). Different environmental factors contribute to the transition of TasA from monomers to well-structured fibres. Native TasA aggregates directly purified from *B. subtilis* cells show a tendency to fibrillate under more acidic conditions and over hydrophobic surfaces [133], which is similar to the tendencies of some of the amyloids presented above. In fact, it has been shown that TasA interacts with synthetic bacterial membranes in vitro, which triggers the polymerization of the protein, influencing the morphology of the fibres [134]. These results show, once again, how environmental conditions can shape the morphology and properties of functional amyloids and how bacteria use the versatile structural and biochemical properties of amyloids to efficiently respond to sudden changes in their surroundings. Moreover, the partially disordered two-domain accessory protein TapA [135] is also a functional amyloid, in which only the first 57 amino acids seem to be important for biofilm architecture [136,137]. This protein has also been detected within the TasA amyloid fibres natively purified from *B. subtilis* cells and is able to self-assemble in vitro into aggregates that display the typical amyloid structure by X-ray diffraction analysis [132]. This finding, along with the fact that TapA enhances the polymerization of TasA in vitro without perturbing the global architecture of the amyloid filament, suggests that both proteins might co-assemble during biofilm formation [132], which would also explain the presence of TapA in TasA fibres isolated from *B. subtilis* cells (Figure 2A). The mechanism by which the two proteins interact has not yet been fully investigated. Recently, molecular dynamics approaches have suggested that the interactions between TasA and TapA rely on key amino acids located in their disordered C-terminal regions, which would explain the high proportion of intrinsic disorder that is present in these regions of both proteins [138].

TasA is also present in the closely related bacilli *Bacillus amyloliquefaciens* and *Bacillus pumilus*, in which the TasA protein has a high proportion of sequence identity in both bacilli [132]. In the group of bacilli that includes *Bacillus cereus* and other related pathogenic bacilli, such as *Bacillus anthracis* or *Bacillus thuringiensis*, *tasA* is part of a genomic region that is analogous to that of *B. subtilis* and is involved in multicellular behaviour and biofilm formation [139]. However, TasA from *B. cereus*, contrary to what occurs in *B. subtilis*, lacks the C-terminal end, which does not affect the protein’s ability to assemble into fibres that morphologically and structurally resemble those found in *B. subtilis* biofilms [132]. The group of *B. cereus* lacks the accessory protein TapA, but another protein called CalY is present in this system; CalY shares an important degree of sequence identity with TasA. CalY is not able to form fibres in vitro and seems to play a complementary role to TasA in the biofilm formation of *B. cereus* [139]. Nonetheless, the CalY protein displays the typical properties of amyloid proteins and has a structural fold similar to that observed for TasA. Interestingly, and similar to TapA, CalY facilitates the polymerization of *B. cereus* TasA without perturbing the structure of the filament, suggesting the possibility of co-assembly [132].

Apart from its structural role in the multicellular behaviour of *Bacillus* spp., these proteins play more complex roles in bacterial physiology. In *B. subtilis*, TasA has a dual role in the physiology of microorganisms: TasA maintains the cell membrane stability and prevents excess cell death under biofilm growth conditions (Figure 2A,B) [140]. Furthermore, during biofilm development, TasA acts as a signal to maintain a subset of motile cells within the bacterial population in a manner that is independent of the biofilm-motility switch, thereby promoting motility and repressing the expression of matrix genes (Figure 2C) [141]. In addition, TasA seems to mediate interactions with different organisms. For instance, this functional amyloid is important for the formation of dual species biofilms between *B. subtilis* and *S. mutans*. The expression of TasA is induced during this interspecies interaction is suggested to contribute to the early stages of the establishment of co-specific adhesion between the two species in a way that seems to be dependent on the presence of dextran, which is a major extracellular polysaccharide present in the extracellular matrix of *S. mutans* [142].

*B. subtilis* is a well-known biocontrol agent and natural enemy of pathogens and is often found in close association with plants [143]. It has been demonstrated that biofilm formation is important for the surface attachment of *B. subtilis* to different plant organs, roots or leaves, favouring antagonist activity [144]. The contribution of TasA to the persistence of *B. subtilis* cells over the plant phylloplane relies on at least three complementary traits: (i) mediating the early stages of the attachment of bacterial cells to the leaves, (ii) structurally ensuring the assembly of the ECM and (iii) preserving the viability of bacterial cells [140,145]. In addition, TasA is required for the attachment, biofilm formation and proliferation of *B. subtilis* in *Arabidopsis thaliana* and tomato roots [146,147]. Indeed, different plant substances, such as polysaccharides and organic acids, are able to trigger the expression of the *tapA* operon, thereby inducing biofilm formation, which would explain the common presence of *B. subtilis* over plant surfaces [146]. In *Bacillus velezensis* QST713, a bacterial strain currently being used to protect the mushroom *Agaricus bisporus* against *Trichoderma aggressivum* f. *europaeum*, another fungal competitor, the expression of TasA and other matrix genes contributes to biofilm formation in compost, which, reciprocally, favours the synthesis of secondary metabolites and antimicrobials [148]. From the *B. cereus* group side, the orthologue of TasA found in *B. cereus* CR4, a strain isolated from activated sludges, shows remarkable bioflocculant activity, which makes this microorganism a good candidate for harvesting microalgae, which is an emerging strategy in the production of biofuels [149]. In *B. thuringiensis*, CalY has been shown to be required for the attachment of bacteria to HeLa cells and for the virulence against the wax moth *Galleria melonella* [150]. However, at the end of the stationary phase, the presence of CalY mostly shifts from the cell surface to the extracellular medium, where the protein is able to polymerize in the form of fibres that support biofilm formation. These results demonstrate the many functionalities that the TasA amyloid system and its orthologues have in the biology and ecology of *Bacillus* spp. and highlight the notion of multiple purposes traditionally attributed to functional amyloids.

## 7. Amyloid Cross-Seeding as a Molecular Crosstalk Mechanism in Bacteria?

Amyloid proteins exhibit a tendency to aggregate, which is one of their defining features. The formation of homologous aggregates and fibres, which are those composed only of one amyloid polypeptide, have been traditionally the most studied. However, the common structural and biochemical properties of amyloid proteins allow for heterologous interactions in a process known as cross-seeding [151]. Cross-seeding and heterologous amyloid aggregates have been well studied in pathological amyloids, such as Aβ_42_, α-synuclein and tau, which are different amyloid peptides that are involved in Alzheimer’s disease [152]. It has been demonstrated in vivo and in vitro how aggregated Aβ_42_ strongly influences tau aggregation and the appearance of tau seeds that are able to trigger the propagation of tau pathology in vivo [153]. Similarly, α-synuclein fibrils are able to induce the formation of neurofibrillary aggregates by cross-seeding with tau intracellularly [154]. 

In contrast, little is known about cross-seeding between different functional bacterial amyloids, and very few cases have been reported in which cross-seeding occurs to serve a specific biological purpose. An example of this has been studied in the gram-negative bacterium *E. coli*. In this microorganism, the curli, which are extracellular proteinaceous filaments involved in biofilm formation and adhesion to different biotic and abiotic surfaces [155,156,157,158], exhibit an amyloid nature. These fibres are composed of two amyloid proteins: CsgA, which is the major curli subunit, and CsgB, which acts as a nucleator, favouring the assembly of curli filaments [8,159]. In vitro experiments have demonstrated that CsgA is a rather promiscuous protein that is able to cross-seed with CsgA orthologues present in other bacterial species (*Salmonella typhimurium*, *Citrobacter koseri* or *Shewanella oneidensis*), even when the sequence identity between these curli orthologues is as low as 30%. Interestingly, this cross-seeding mechanism is also evidenced in the interaction between *E. coli* and *S. typhimurium*, where the CsgA subunit produced and secreted by *E. coli* can complement the defects of a *csgA* mutant of *S. typhimurium* and vice versa. Similar results were also obtained in the study of the CsgB subunit. This complementation restores the adhesion capacity of the mixed community and promotes dual-species biofilms between the two bacteria, demonstrating how amyloid cross-seeding can modulate not only intraspecies but also interspecies microbial physiology and ecology [160].

A number of experimental studies also suggest the existence of intra- and interspecies complementation of functional amyloids of *Bacillus*. First, biofilm formation is restored in mixed cultures of *eps* and *tasA* isogenic single mutants, suggesting a structural role of TasA provided by the *eps* mutant for the benefit of the community [130]. However, unlike curli of gram-negative bacteria, it seems more likely that the two subunits involved in TasA amyloid fibre formation (TapA and TasA) have to be produced by the same cell to fulfil their functionality, and this is supported by two complementary experimental studies: (i) the co-culture of *tasA* and *tapA* does not produce wrinkled colonies typical of a WT biofilm [130], and (ii) purified TasA does not rescue biofilm formation in a *tapA* mutant [132]. The precise mechanism that drives the collaboration of the two proteins in fibre assembly remains to be determined. Interestingly, and in agreement with this finding of intraspecies interaction, the chromosomal region containing TasA and CalY in *B. cereus* rescues biofilm formation in a *B. subtilis* mutant for the whole *tapA* operon. Similarly, and more specifically, *B. cereus* TasA, when heterologously expressed in *B. subtilis*, was able to cause the same reversion. The formation of TasA fibres in both *B. subtilis* backgrounds and the Congo red staining of the colony were indicative of the preservation of the amyloid nature of *B. cereus* TasA expressed in *B. subtilis* [139]. Further structural and functional studies with purified proteins from both species will shed light on the exact molecular mechanism of interspecies complementation.

In the gram-positive bacterium *S. aureus*, the PSMs described in the above section are involved in biofilm formation [11,79]. Recent studies suggest that each individual PSM exhibits a unique behaviour in terms of aggregation and amyloid formation. Moreover, it has been shown that, in vitro, PSMs have the ability to interact with one another by cross-seeding, in which preformed seeds of one PSM are able to induce polymerization of other PSMs. In fact, PSMs exhibit a certain degree of specificity in their cross-seeding ability; for instance, PSMα3 seeds promote only self-aggregation and aggregation of PSMα1, PSMβ1 is able to cross-seed only with PSMα1 and PSMβ2, and PSMβ2 seeds accelerate only the aggregation of PSMα1, with some mild effects on PSMβ1, PSMα2 and the δ toxin. PSMα1 seems to be less strict in its ability to cross-seed than the other PSMs and can induce aggregation in all of them [161]. Overall, these results suggest a possible interplay of all these peptides in the multicellular behaviour of *S. aureus*; however, more experimental evidence is needed to associate a biological function with this cross-seeding activity in the context of biofilm formation.

## 8. Concluding Remarks

In bacteria, research efforts have focused on the role of functional amyloids as structural scaffolds within the biofilms, however, additional roles in bacterial physiology and interaction with hosts are currently being discovered. In Gram-positive bacteria, there is a vast diversity of amyloid systems, each one with its own protein components and peculiarities in the process of polymerization. This biochemical variety increases the chances of finding new functions associated with amyloid proteins or the amyloid systems themselves. The tendency shown by these proteins to interact with each other in different biological contexts makes cross-seeding a potential mechanism for the diversification of communication between different species and therefore the structure of heterogenous microbial communities. This biological promiscuity can also be translated to the interaction with eucaryotes, where amyloids may induce or repress the polymerization of their eucaryotic siblings, thus introducing different responses at a cellular, immunological or physiological level. These are exciting questions that will provide us with new and undescribed roles of microbial amyloids, knowledge that will additionally help understanding of the real impact of this family of proteins in the ecology of gram-positive bacteria.

## Figures and Tables

**Figure 1 microorganisms-08-02020-f001:**
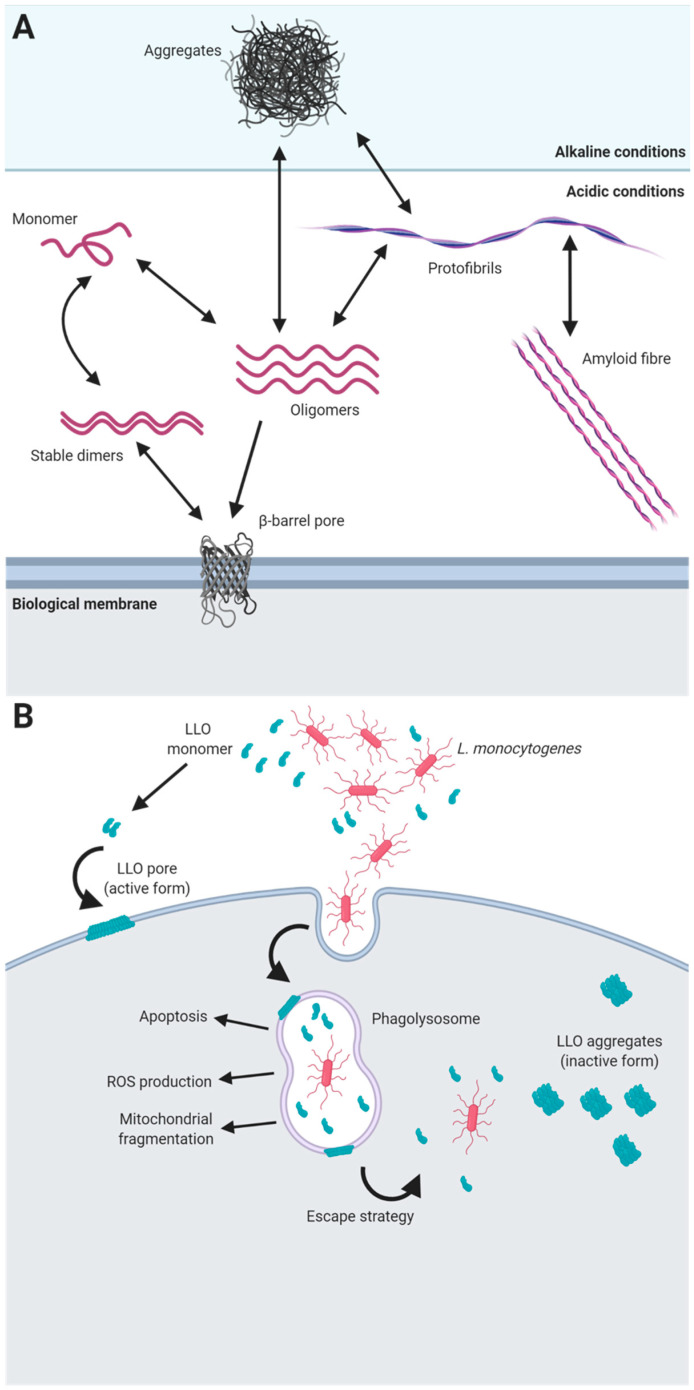
The role of functional amyloids as toxins. (**A**) Environmental conditions influence the aggregation state and kinetics of amyloid proteins. The folding of amyloid proteins into different structures is modulated by environmental conditions, i.e., pH, as exemplified in this figure. Protein monomers are able to form stable dimers that adopt a beta-hairpin structure, which in contact with biological membranes switch to a β-barrel pore. Furthermore, monomers are able to assemble into oligomers that finally give rise to mature amyloid fibres. At non optimal conditions, misfolded monomers or oligomers tend to form amorphous aggregates that finally can lead to the formation of protofibrils or β-barrel pores when appropriate conditions are met. (**B**) Mechanism of action of LLO toxin of *L. monocytogenes*. The bacteria secretes monomers of LLO that form stable dimers that are able to oligomerize building pores through the interaction with biological membranes. Inside of the phagolysosome, the toxin is secreted by the bacteria, limiting ROS production and causing the lysis of the lysosome, which contributes to the survival of the pathogen. LLO is also implicated in apoptosis and mitochondrial fragmentation. At alkaline conditions, such as the ones found in the cytoplasm, LLO monomers are unable to polymerize and tend to form amyloid aggregates which inactivate the cytotoxic activity of the protein [125].

**Figure 2 microorganisms-08-02020-f002:**
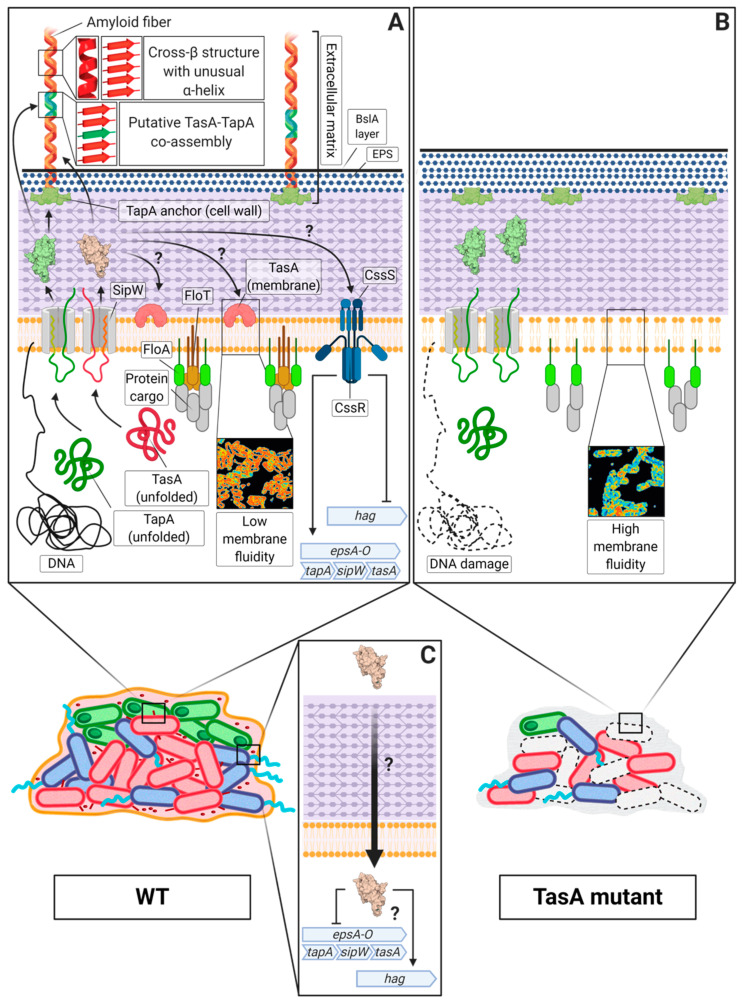
The TasA system in *B. subtilis*. The figure depicts the current knowledge regarding the TasA system in *B. subtilis* by showing two situations in the presence or absence of TasA. Biofilm formation requires the matrix proteins TasA and TapA, that are both essential for extracellular matrix production (ECM) and other cell functions, and the signal peptidase SipW. The biofilm schemes shown in the lower part of the figure include ECM producers (red), motile cells (blue), sporulating cells (green) and dead cells (dashed). (**A**). The TasA amyloid fibres provide structure to the ECM and exhibit a cross-β structure with a particular proportion of α –helix. These fibres are anchored to the cell surface by TapA, which also promotes TasA fibrillation and can be found as part of the fibres. In addition, TasA is also found in the cell membrane, where it maintains membrane stability, hypothetically, in proximity to functional membrane microdomains [140]. Under biofilm-inducing conditions, the expression of the matrix genes is maintained by the biofilm motility switch, which it has been recently reported to include the two-component response regulator CssS/CssR, that might sense TasA repressing the expression of flagellar genes [141]. (**B**). The absence of TasA leads to an increase of the cell death within the bacterial population, caused by a destabilization of the cell membrane and the subsequent alteration of membrane dynamics, which leads to the mislocalization of the flotillin FloT and other cellular disturbances such as DNA damage [140]. (**C**). In addition, TasA promotes motility by an unknown mechanism in a subset of the biofilm cells that allow for the proper expansion of the colony [141].

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
