# Peer review of "Multifunctional Amyloids in the Biology of Gram-Positive Bacteria"

_microorganisms, 2020, doi:10.3390/microorganisms8122020_

Round 1

Reviewer 1 Report

The review report was done by Diego Romero group “Multifunctional amyloids in the biology of gram-positive bacteria” is a good research review to study functional amyloids in gram-positive bacteria which are diverse in their assembly mechanisms and unusually specific in their biological functions. Authors documented cross-seeding between functional amyloids as an emerging theme in interspecies interactions that contributes to the diversification of bacterial biology.

This review has shown previous literature on how some amyloids and their ability to interact with one another in a process known as cross-seeding.

This review is a simple and nice document that shows multifunctional amyloids in the biology of gram-positive bacteria. The major advantage of the present documented report is that the latest findings of different bacterial amyloid systems, with particular emphasis on gram-positive bacteria, their contribution to bacterial multicellularity and microbe-host interactions. There are not many literature reviews, similar to a present documented report. Given the importance of practicality for this work, I recommend the publication of this manuscript in the microorganisms.

Author Response

We do really appreciate and are very glad on the comments done by the reviewer on our manuscript. 

In response to the English language, we do, on regular bases, send our manuscript to expert native English editorial before submission to publication.

Reviewer 2 Report

The authors examine the multi-functionality of many bacterial amyloids, focusing specifically on those produced by Gram-positive bacteria. After a brief introduction of functional amyloids, the review discusses the formation and biological roles of several well-studied bacterial amyloids. This includes chaplins, PSM’s, adhesins, LLO, and the TasA amyloid system. The authors highlighting how functional amyloids perform various tasks for their bacteria and in some cases aid in symbiotic relationships. The manuscript closes with a discussion of cross-seeding, a biological phenomenon that has been more clearly described in pathological amyloids and remains relatively unexplored in bacterial amyloids.

There are a few minor points the authors should address before the manuscript is recommended for publication:

  • The sentence on line 46-49 could probably be split into two sentences.
  • Reference #34 seems out of place. It could be replaced with the following reference:
    • Nguyen, P. Q., Botyanszki, Z., Tay, P. K. R., and Joshi, N. S. (2014). Programmable biofilm-based materials from engineered curli nanofibres. Nat. Commun.5:5945. doi: 10.1038/ncomms5945
    • In general, the authors choose to provide references for related sentences together at the end of a section or paragraph. While this is fine, they might decide to provide references for the particular sentences or statements as they occur. That way it is clear to the reader where the work can be found.
  • The paragraph starting on line 55 should include a different topic sentence.
  • The sentence on line 114-118 should include reference 37.
  • I don’t see any of the figures that are referenced in the text. Not sure if I just missed them or if they were inadvertently excluded from the upload.
  • The review would benefit from a concluding paragraph that pulls together the important thesis that the authors would like to trumpet.
  • The connection between Bap and Listeria is somewhat unsupported and the authors may choose to leave this out.

Author Response

We appreciate the reviewer’s comments and the time invested in carefully reading the manuscript. The minor points raised by the reviewer are addressed below.

  • The sentence on line 46-49 could probably be split into two sentences.

This has been corrected. A full stop has been added at the end of the first sentence (line 63) and the new sentence begins in line 64.

  • Reference #34 seems out of place. It could be replaced with the following reference: Nguyen, P. Q., Botyanszki, Z., Tay, P. K. R., and Joshi, N. S. (2014). Programmable biofilm-based materials from engineered curli nanofibres. Nat. Commun.5:5945. doi: 10.1038/ncomms5945

Due to the addition of new references to the manuscript (by suggestions from other reviewers) the order of the references has changed. The reference #34 (now reference #39) mentioned by the reviewer:

Astorga, F., et al., Distributional ecology of Andes hantavirus: a macroecological approach. Int. J. Health Geogr.

has been substituted by

Nguyen, P.Q., et al., Programmable biofilm-based materials from engineered curli nanofibres. Nat Commun, 2014. 5: p. 4945.

  • In general, the authors choose to provide references for related sentences together at the end of a section or paragraph. While this is fine, they might decide to provide references for the particular sentences or statements as they occur. That way it is clear to the reader where the work can be found.

We appreciate the reviewer’s suggestion and we will consider it for future manuscripts.

  • The paragraph starting on line 55 should include a different topic sentence.

We have maintained the same topic sentence; however, we have split the paragraph (line 60) to separate two clearly defined topics.

  • The sentence on line 114-118 should include reference 37.

This has been corrected. In addition, we found that the reference mentioned by the reviewer (Elliot, M.A., et al., The chaplins: a family of hydrophobic cell-surface proteins involved in aerial mycelium formation in Streptomyces coelicolor. Genes Dev, 2003. 17) was duplicated. This has been corrected as well. The reference is now reference #28

  • I don’t see any of the figures that are referenced in the text. Not sure if I just missed them or if they were inadvertently excluded from the upload.

We apologize for the mistake. The two figures have been uploaded now.

  • The review would benefit from a concluding paragraph that pulls together the important thesis that the authors would like to trumpet.

We agree with the reviewer. A “Concluding remarks” sections has been added at the end of the manuscript prior to the “References” section.

  • The connection between Bap and Listeria is somewhat unsupported and the authors may choose to leave this out.

Lines 292-298 from the original manuscript have been deleted. Only the mention of the locus Lmo0435, designated in the literature as BapL, and the similarity between this and other Bap-type proteins that do exhibit amyloid properties (as described in the literature) is mentioned (lines 303-306 of the new manuscript).

Reviewer 3 Report

This review article comprehensively describes multifunctional amyloid associated with gram-positive bacteria. Overall, this manuscript is well written.

1) The current review describes the gram-positive bacteria. However, several reports proposed the amyloid of gram-negative bacteria such as E. coli (Tonomuta, Journal of Cerebral Blood Flow & Metabolism, 2020.  DOI: 10.1177/0271678X20918031). Please add the description of amyloid associated with gram-negative bacteria or introduce an appropriate review article about it for readers.

2) In this manuscript, S. mutans is introduced as a causative agent of infectious endocarditis. Cnm protein of S. mutans is reported to be associated with the pathogenesis of infectious endocarditis (Sci Rep. 2020 Nov 5;10(1):19118. doi: 10.1038/s41598-020-75933-6.). Please mention it.

3) Regarding amyloid cross-seeding, the authors describe the propagation of Aβ42 and tau. It is well known that α-Synuclein possesses cross-seeding property. Considering that both Alzheimer’s disease and Parkinson’s disease are introduced as protein misfolding diseases in introduction, cross-seeding property of α-Synuclein should be mentioned.

4) The fact that amyloid-β pathology was transmitted in human is important in this field (Nature volume 525, pages247–250(2015)). Please describe it.

5) There is no table or figure. In order to promote better understanding, it would be better to add a table or figure, which summarize the description.

Author Response

This review article comprehensively describes multifunctional amyloid associated with gram-positive bacteria. Overall, this manuscript is well written.

We express our gratitude to the reviewer for reading the article and giving its opinion. The minor points raised by the reviewer are addressed below.

  • The current review describes the gram-positive bacteria. However, several reports proposed the amyloid of gram-negative bacteria such as coli (Tonomuta, Journal of Cerebral Blood Flow & Metabolism, 2020.  DOI: 10.1177/0271678X20918031). Please add the description of amyloid associated with gram-negative bacteria or introduce an appropriate review article about it for readers.

The main topic of this review article is amyloids in gram-positive bacteria, as we believe it is a more unexplored field compared to the curli system present in E. coli and other gram-negative bacteria. In any case, we added some references in the introduction related to both groups of bacteria in line 44, where the readers can find more information about the different roles of amyloid proteins in gram-negative bacteria.

  • Sgro, G.G., et al., Contribution of a harpin protein from Xanthomonas axonopodis pv. citri to pathogen virulence.Mol Plant Pathol, 2012. 13(9): p. 1047-59.
  • Shahnawaz, M. and C. Soto, Microcin amyloid fibrils A are reservoir of toxic oligomeric species. J Biol Chem, 2012. 287(15): p. 11665-76.
  • Chapman, M.R., et al., Role of Escherichia coli curli operons in directing amyloid fiber formation. Science, 2002. 295(5556): p. 851-5.
  • Dueholm, M.S., et al., Functional amyloid in Pseudomonas. Mol Microbiol, 2010. 77(4): p. 1009-20.
  • Schwartz, K., et al., Functional amyloids composed of phenol soluble modulins stabilize Staphylococcus aureus biofilms. PLoS Pathog, 2012. 8(6): p. e1002744.
  • Gibson, D.L., et al., AgfC and AgfE facilitate extracellular thin aggregative fimbriae synthesis in Salmonella enteritidis. Microbiology (Reading), 2007. 153(Pt 4): p. 1131-1140.
  • Elliot, M.A., et al., The chaplins: a family of hydrophobic cell-surface proteins involved in aerial mycelium formation in Streptomyces coelicolor. Genes Dev, 2003. 17(14): p. 1727-40.
  • Oli, M.W., et al., Functional amyloid formation by Streptococcus mutans. Microbiology (Reading), 2012. 158(Pt 12): p. 2903-2916.
  • Oh, J., et al., Amyloidogenesis of type III-dependent harpins from plant pathogenic bacteria. J Biol Chem, 2007. 282(18): p. 13601-9.
  • Molina-Garcia, L., et al., Functional amyloids as inhibitors of plasmid DNA replication. Sci Rep, 2016. 6: p. 25425.
  • Rouse, S.L., S.J. Matthews, and M.S. Dueholm, Ecology and Biogenesis of Functional Amyloids in Pseudomonas.J Mol Biol, 2018. 430(20): p. 3685-3695.
  • Salinas, N., et al., Emerging Roles of Functional Bacterial Amyloids in Gene Regulation, Toxicity, and Immunomodulation. Microbiol Mol Biol Rev, 2020. 85(1).
  • In this manuscript, mutans is introduced as a causative agent of infectious endocarditis. Cnm protein of S. mutans is reported to be associated with the pathogenesis of infectious endocarditis (Sci Rep. 2020 Nov 5;10(1):19118. doi: 10.1038/s41598-020-75933-6.). Please mention it.

This review is focused on the various roles of amyloid proteins. The Cnm protein of S. mutants is a collagen-binding adhesin, according to the literature (Infect Immun. 2015;83(5):2001-2010. doi:10.1128/IAI.03022-14; J Bacteriol. 2014;196(15):2789-2797. doi:10.1128/JB.01783-14). However, we have found no evidence regarding the amyloid nature of Cnm. Therefore, we have opted for not including a description of this protein in the manuscript.   

Regarding amyloid cross-seeding, the authors describe the propagation of Aβ42 and tau. It is well known that α-Synuclein possesses cross-seeding property. Considering that both Alzheimer’s disease and Parkinson’s disease are introduced as protein misfolding diseases in introduction, cross-seeding property of α-Synuclein should be mentioned.

We have added information regarding the cross-seeding process between Tau and α-synuclein and its involvement in Alzheimer’s disease (lines 426-427).

  • The fact that amyloid-β pathology was transmitted in human is important in this field (Nature volume 525, pages247–250(2015)). Please describe it.

This information has been added to the introduction (lines 37-41), describing the transmission phenomenon of the Aβ peptide and Alzheimer’s disease between humans.

  • There is no table or figure. In order to promote better understanding, it would be better to add a table or figure, which summarize the description.

We apologize for the mistake. The two figures have been uploaded now.